# Is deeper better? It depends on locality of relevant features

## Abstract

It has been recognized that a heavily overparameterized artificial neural network exhibits surprisingly good generalization performance in various machine-learning tasks. Recent theoretical studies have made attempts to unveil the mystery of the overparameterization. In most of those previous works, the overparameterization is achieved by increasing the width of the network, while the effect of increasing the depth has remained less well understood. In this work, we investigate the effect of increasing the depth within an overparameterized regime. To gain an insight into the advantage of depth, we introduce local and global labels as abstract but simple classification rules. It turns out that the locality of the relevant feature for a given classification rule plays a key role; our experimental results suggest that deeper is better for local labels, whereas shallower is better for global labels. We also compare the results of finite networks with those of the neural tangent kernel (NTK), which is equivalent to an infinitely wide network with a proper initialization and an infinitesimal learning rate. It is shown that the NTK does not correctly capture the depth dependence of the generalization performance, which indicates the importance of the feature learning, rather than the lazy learning.

## 1 Introduction

Deep learning has achieved unparalleled success in various tasks of artificial intelligence such as image classification (Krizhevsky et al., 2012; LeCun et al., 2015) and speech recognition (Hinton et al., 2012). Remarkably, in modern machine learning applications, impressive generalization performance has been observed in an *overparameterized* regime, in which the number of parameters in the network is much larger than that of training data samples. Contrary to what we learn in classical learning theory, an overparameterized network fits random labels and yet generalizes very well without serious overfitting (Zhang et al., 2017). We do not have general theory that explains why deep learning works so well.

Recently, the learning dynamics and the generalization power of heavily overparameterized wide neural networks have extensively been studied. It has been reported that training of an overparameterized network easily achieves zero training error without getting stuck in local minima of the loss landscape (Zhang et al., 2017; Baity-Jesi et al., 2018). Mathematically rigorous results have also been obtained (Allen-Zhu et al., 2019; Du et al., 2019). From a different perspective, theory of the neural tangent kernel (NTK) has been developed as a new tool to investigate an overparameterized network with an infinite width (Jacot et al., 2018; Arora et al., 2019), which simply explains the reason why a sufficiently wide neural network can achieve a global minimum of the training loss.

As for generalization, a "double-descent" phenomenon has attracted much attention (Spigler et al., 2019; Belkin et al., 2019). The standard bias-variance tradeoff scenario predicts a U-shaped curve of the test error (Geman et al., 1992); however, one finds the double-descent curve, which implies that an increased model capacity beyond the interpolation threshold results in improved performance. This finding triggered detailed studies on the behavior of the bias and variance in an overparameterized regime (Neal et al., 2019; D'Ascoli et al., 2020). The double-descent phenomenon is not explained by traditional complexity measures such as the Vapnik-Chervonenkis dimension and the Rademacher complexity (Mohri et al., 2018), and hence one seeks for new complexity measures of deep neural networks that can prove better generalization bounds (Dziugaite & Roy, 2017; Neyshabur et al., 2017; 2019; Arora et al., 2018; Nagarajan & Kolter, 2017; Pérez et al., 2019).

These theoretical efforts mainly focus on the effect of increasing the network width, but benefits of the network depth remain unclear. It is known that expressivity of a deep neural network grows exponentially with the depth rather than the width (Poole et al., 2016). See also Bianchini & Scarselli (2014); Montúfar et al. (2014). However, it is unclear whether exponential expressivity does lead to better generalization (Ba & Caruana, 2014; Becker et al., 2020). It is also nontrivial whether typical problems encountered in practice require such high expressivity. Although some works (Eldan & Shamir, 2016; Safran & Shamir, 2017) have shown that there exist simple and natural functions that are efficiently approximated by a network with two hidden layers but not by a network with one hidden layer, a recent work (Malach & Shalev-Shwartz, 2019) has demonstrated that a deep network can only learn functions that are well approximated by a shallow network by using a gradient-based optimization algorithm, which indicates that benefits of the depth are not due to high expressivity of deep networks. Some other recent works have reported no clear advantage of the depth in an overparameterized regime (Geiger et al., 2019a;b).

To gain an insight into the advantage of the depth, in the present paper, we report our experimental study on the depth and width dependences of generalization in abstract but simple, well-controlled classification tasks with fully connected neural networks. We introduce *local labels* and *global labels*, both of which give simple mappings between inputs and output class labels. By "local", we mean that the label is determined only by a few components of the input vector. On the other hand, a global label is determined by a global feature that is expressed as a sum of local quantities, and hence all components of an input contribute to the global label. Our experiments show strong depth-dependences of the generalization error for those simple input-output mappings. In particular, we find that *deeper is better for local labels, while shallower is better for global labels*. This result implies that the depth is not always advantageous, but the locality of relevant features would give us a clue for understanding the advantage the depth brings about.

We also compare the generalization performance of a trained network of a finite width with that of the kernel method with the NTK. The latter corresponds to the infinite-width limit of a fully connected network with an appropriate initialization and an infinitesimal learning rate (Jacot et al., 2018), which is referred to as the NTK limit. In the NTK limit, the network parameters stay close to their initial values during training, which is called the *lazy learning* (Chizat et al., 2019). It is known that a wide but finite network can still be in the lazy learning regime for sufficiently small learning rates (Ji & Telgarsky, 2019; Chen et al., 2019). We however find that even if the width increases, in some cases the generalization error *at an optimal learning rate* does not converge to the NTK limit. In such a case, a finite-width network shows much better generalization compared with the kernel learning with the NTK. This finding indicates the importance of the *feature learning*, in which network parameters change to learn relevant features.

## 2 SETTING

We consider a classification task with a training dataset $\mathcal{D} = \{(x^{(\mu)}, y^{(\mu)}) : \mu = 1, 2, \ldots, N\}$, where $x^{(\mu)} \in \mathbb{R}^d$ is an input data and $y^{(\mu)} \in \{1, 2, \ldots, K\}$ is its label. In this work, we consider the binary classification, $K = 2$, unless otherwise stated.

### 2.1 DATASET

Each input $x = (x_1, x_2, \ldots, x_d)^{\mathrm{T}}$ is a $d$-dimensional vector taken from i.i.d. Gaussian random variables of zero mean and unit variance, where $a^{\mathrm{T}}$ is the transpose of vector $a$. For each input $x$, we assign a label $y$ according to one of the following rules.

### $k$-LOCAL LABEL

We randomly fix integers $\{i_1, i_2, \ldots, i_k\}$ with $1 \leq i_1 < i_2 < \cdots < i_k \leq d$. In the "$k$-local" label, the relevant feature is given by the product of the $k$ components of an input $x$, that is, the label $y$ is determined by

$$y = \begin{cases} 1 & \text{if } x_{i_1} x_{i_2} \ldots x_{i_k} \geq 0; \\ 2 & \text{otherwise.} \end{cases} \tag{1}$$

This label is said to be local in the sense that $y$ is completely determined by just the $k$ components of an input $x$.[1] For fully connected networks considered in this paper, without loss of generality, we can choose $i_1 = 1$, $i_2 = 2$,... $i_k = k$ because of the permutation symmetry with respect to indices of input vectors.

$k$-GLOBAL LABEL

We again fix $1 \leq i_1 < i_2 < \cdots < i_k \leq d$. Let us define

$$M = \sum_{j=1}^{d} x_{j+i_1} x_{j+i_2} \ldots x_{j+i_k}, \tag{2}$$

where the convention $x_{d+i} = x_i$ is used. The $k$-global label $y$ for $x$ is defined by

$$y = \begin{cases} 1 & \text{if } M \geq 0; \\ 2 & \text{otherwise.} \end{cases} \tag{3}$$

The relevant feature $M$ for this label is given by a uniform sum of the product of $k$ components of the input vector. Every component of $x$ contributes to this "$k$-global" label, in contrast to the $k$-local label with $k < d$.

## 2.2 NETWORK ARCHITECTURE

In the present work, we consider fully connected feedforward neural networks with $L$ hidden layers of width $H$. We call $L$ and $H$ the depth and the width of the network, respectively. The output of the network $f(x)$ for an input vector $x \in \mathbb{R}^d$ is determined as follows:

$$\begin{cases} f(x) = z^{(L+1)} = w^{(L+1)} z^{(L)} + b^{(L+1)}; \\ z^{(l)} = \varphi \left( w^{(l)} z^{(l-1)} + b^{(l)} \right) \text{ for } l = 1, 2, \ldots, L; \\ z^{(0)} = x, \end{cases} \tag{4}$$

where $\varphi(x) = \max\{x, 0\}$ is the component-wise ReLU activation function, $z^{(l)}$ is the output of the $l$th layer, and

$$w^{(l)} \in \begin{cases} \mathbb{R}^{K \times H} \text{ for } l = L + 1; \\ \mathbb{R}^{H \times H} \text{ for } l = 2, 3, \ldots, L; \\ \mathbb{R}^{H \times d} \text{ for } l = 1, \end{cases} \qquad b^{(l)} \in \begin{cases} \mathbb{R}^{K} \text{ for } l = L + 1; \\ \mathbb{R}^{H} \text{ for } l = 1, 2, \ldots, L \end{cases} \tag{5}$$

are the weights and the biases, respectively. Let us denote by $w$ the set of all the weights and biases in the network. We focus on an overparameterized regime, where the number of network parameters (the number of components of $w$) exceeds $N$, the number of training data points.

## 2.3 SUPERVISED LEARNING

The network parameters $w$ are adjusted to correctly classify the training data. It is done by minimizing the softmax cross-entropy loss $L(w)$ given by

$$L(w) = \frac{1}{N} \sum_{\mu=1}^{N} \ell \left( f(x^{(\mu)}), y^{(\mu)} \right), \quad \ell \left( f(x), y \right) = -\ln \frac{e^{f_y(x)}}{\sum_{i=1}^{K} e^{f_i(x)}} = -f_y(x) + \ln \sum_{i=1}^{K} e^{f_i(x)}, \tag{6}$$

where the $i$th component of $f(x)$ is denoted by $f_i(x)$.

The training of the network is done by the stochastic gradient descent (SGD) with learning rate $\eta$ and the mini-batch size $B$. That is, for each mini-batch $\mathcal{B} \subset \mathcal{D}$ with $|\mathcal{B}| = B$, the network parameter

---

[1]The locality here does not necessarily imply that $k$ points $i_1, i_2, \ldots, i_k$ are spatially close to each other. Such a usage of the terminology "$k$-local" has been found (but $k$-global is also called "$k$-local") in the field of quantum information (Kempe et al., 2006).

$w_t$ at time $t$ is updated as

$$w_{t+1} = w_t - \eta \nabla_w L_B(w), \qquad L_B(w) = \frac{1}{B} \sum_{\mu \in \mathcal{B}} \ell \left( f(x^{(\mu)}), y^{(\mu)} \right). \qquad (7)$$

Throughout the paper, we fix $B = 50$. Meanwhile, we optimize $\eta > 0$ before training (we explain the detail later). Biases are initialized to be zero, and weights are initialized using the Glorot initialization (Glorot & Bengio, 2010).[2]

The trained network classifies an input $x_\mu$ to the class $\hat{y}^{(\mu)} = \arg\max_{i \in \{1,2,\ldots,K\}} f_i(x^{(\mu)})$. Let us then define the training error as

$$\mathcal{E}_{\text{train}} = \frac{1}{N} \sum_{\mu=1}^{N} \left( 1 - \delta_{y^{(\mu)}, \hat{y}^{(\mu)}} \right), \qquad (8)$$

that is the miss-classification rate for the training data $\mathcal{D}$. We train our network until $\mathcal{E}_{\text{train}} = 0$ is achieved, i.e., all the training data samples are correctly classified, which is possible in an overparameterized regime.

For a training dataset $\mathcal{D}$, we first perform the 10-fold cross validation to optimize the learning rate $\eta$ under the Bayesian optimization method (Snoek et al., 2012), and then perform the training via the SGD by using the full training dataset. In the optimization of $\eta$, we try to minimize the miss-classification ratio for the validation data.

The generalization performance of a trained network is measured by computing the test error. We prepare the test data $\mathcal{D}_{\text{test}} = \{(x'^{(\mu)}, y'^{(\mu)}) : \mu = 1, 2, \ldots, N_{\text{test}}\}$ independently from the training data $\mathcal{D}$. The test error $\mathcal{E}_{\text{test}}$ is defined as the miss-classification ratio for $\mathcal{D}_{\text{test}}$, i,.e.,

$$\mathcal{E}_{\text{test}} = \frac{1}{N_{\text{test}}} \sum_{\mu=1}^{N_{\text{test}}} \left( 1 - \delta_{y'^{(\mu)}, \hat{y}'^{(\mu)}} \right), \qquad (9)$$

where $\hat{y}'^{(\mu)} = \arg\max_i f_i(x'^{(\mu)})$ is the prediction of our trained network. In our experiment discussed in Sec. 3, we fix $N_{\text{test}} = 10^5$.

## 2.4 NEURAL TANGENT KERNEL

Following Arora et al. (2019) and Cao & Gu (2019), let us consider a network of depth $L$ and width $H$ whose biases $\{b^{(l)}\}$ and weights $\{w^{(l)}\}$ are randomly initialized as $b_i^{(l)} = \beta \tilde{b}_i^{(l)}$ with $\tilde{b}_i^{(l)} \sim \mathcal{N}(0, 1)$ and $w_{ij}^{(l)} = \sqrt{2/n_{l-1}} \tilde{w}_{ij}^{(l)}$ with $\tilde{w}_{ij}^{(l)} \sim \mathcal{N}(0, 1)$ for every $l$, where $n_l$ is the number of neurons in the $l$th layer, i.e., $n_0 = d$, $n_1 = n_2 = \cdots = n_L = H$. Let us denote by $\tilde{w}$ the set of all the scaled weights $\{\tilde{w}^{(l)}\}$ and biases $\{\tilde{b}^{(l)}\}$. The network output is written as $f(x, \tilde{w})$. When the network is sufficiently wide and the learning rate is sufficiently small, the network parameters $\tilde{w}$ stay close to their randomly initialized values $\tilde{w}_0$ during training, and hence $f(x, \tilde{w})$ is approximated by a linear function of $\tilde{w} - \tilde{w}_0$: $f(x, \tilde{w}) = f(x, \tilde{w}_0) + \nabla_{\tilde{w}} f(x, \tilde{w})|_{\tilde{w}=\tilde{w}_0} \cdot (\tilde{w} - \tilde{w}_0)$. As a result, the minimization of the mean-squared error $L_{\text{MSE}} = (1/N) \sum_{\mu=1}^{N} [f(x^{(\mu)}, \tilde{w}) - \vec{y}^{(\mu)}]^2$, where $\vec{y}^{(\mu)} \in \mathbb{R}^K$ is the one-hot representation of the label $y^{(\mu)}$, is equivalent to the kernel regression with the NTK $\Theta_{ij}^{(L)}(x, x')$ $(i, j = 1, 2, \ldots, K)$ defined as

$$\Theta_{ij}^{(L)}(x, x') = \lim_{H \to \infty} \mathbb{E}_{\tilde{w}} \left[ (\nabla_{\tilde{w}} f_i(x, \tilde{w}))^{\mathrm{T}} (\nabla_{\tilde{w}} f_j(x, \tilde{w})) \right], \qquad (10)$$

where $\mathbb{E}_{\tilde{w}}$ denotes the average over random initializations of $\tilde{w}$ (Jacot et al., 2018). The parameter $\beta$ controls the impact of bias terms, and we set $\beta = 0.1$ in our numerical experiment following Jacot et al. (2018). By using the ReLU activation function, we can give an explicit expression of the NTK that is suited for numerical calculations. Such formulas are given in Appendix A.

---

[2]We also tried the He initialization (He et al., 2015) and confirmed that results are similar to the ones obtained by the Glorot initialization, in particular when input vectors are normalized as $\|x\| = 1$.

(a) 1-local                                    (b) 1-global

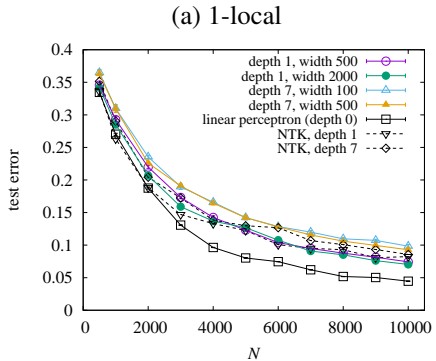
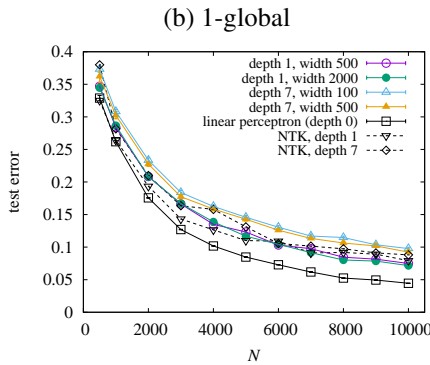

Figure 1: Test error against the number of training data samples $N$ for several network architectures specified by the depth and width for (a) the 1-local label and (b) the 1-global label. Test errors calculated by the NTK of depth 1 and 7 are also plotted. Error bars are smaller than the symbols.

It is shown that the NTK takes the form $\Theta_{ij}^{(L)}(x, x') = \delta_{i,j}\Theta^{(L)}(x, x')$, and the minimization of the mean-squared error with an infinitesimal weight decay yields the output function

$$f^{\mathrm{NTK}}(x) = \sum_{\mu,\nu=1}^{N} \Theta^{(L)}(x, x^{(\mu)})\left(K^{-1}\right)_{\mu\nu}\vec{y}^{(\nu)}, \tag{11}$$

where $K^{-1}$ is the inverse matrix of the Gram matrix $K_{\mu\nu} = \Theta^{(L)}(x^{(\mu)}, x^{(\nu)})$. An input data $x$ is classified to $\hat{y} = \arg\max_{i\in\{1,2,\dots,K\}} f_i^{\mathrm{NTK}}(x)$.

## 3 EXPERIMENTAL RESULTS

We now present our experimental results. For each data point, the training dataset $\mathcal{D}$ is fixed and we optimize the learning rate $\eta$ via the 10-fold cross validation with the Bayesian optimization method (we used the package provided in Nogueira (2014)). We used the optimized $\eta$ to train our network. At every 50 epochs we compute the training error $\mathcal{E}_{\mathrm{train}}$, and we stop the training if $\mathcal{E}_{\mathrm{train}} = 0$. For the fixed dataset $\mathcal{D}$ and the optimized learning rate $\eta$, the training is performed 10 times and calculate the average and the standard deviation of test errors $\mathcal{E}_{\mathrm{test}}$.

We present experimental results for the softmax cross-entropy loss in this section and for the mean-square loss in Appendix D. Our main result holds for both cases, although the choice of the loss function quantitatively affects the generalization performance.

### 3.1 1-LOCAL AND 1-GLOBAL LABELS

In the 1-local and 1-global labels, the relevant feature is a linear function of the input vector. Therefore, in principle, even a linear network can correctly classify the data. Figure 1 shows the generalization errors in nonlinear networks of the varying depth and width as well as those in the linear perceptron (the network of zero depth). The input dimension is set to be $d = 1000$. We also plotted test errors calculated by the NTK, but we postpone the discussion about the NTK until Sec. 3.3.

Figure 1 shows that in both 1-local and 1-global labels, the test error decreases with the network width, and a shallower network ($L = 1$) shows better generalization compared with a deeper one ($L = 7$). The linear perceptron shows the best generalization performance, which is natural because it is the simplest network that is capable of learning the relevant feature associated with the 1-local or 1-global label. Remarkably, test errors of nonlinear networks ($L = 1$ and $L = 7$) are not so large compared with those of the linear perceptron, although nonlinear networks are much more complex than the linear perceptron.

For a given network architecture, we do not see any important difference between the results for 1-local and 1-global labels, which would be explained by the fact that these labels are transformed to each other via the Fourier transformation of input vectors.

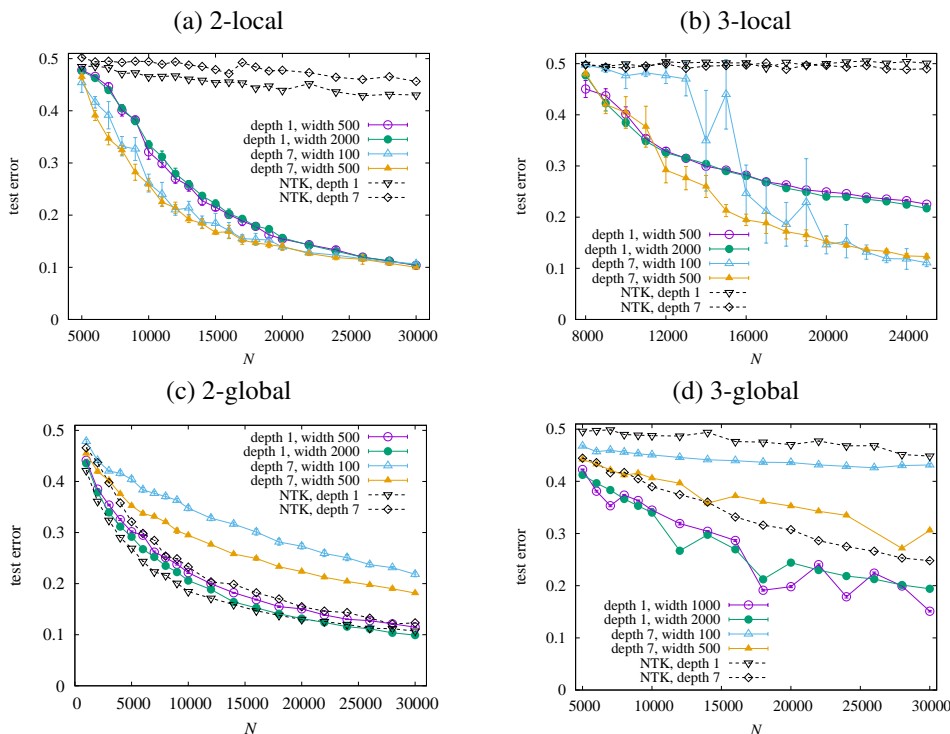

Figure 2: Test error against the number of training data samples $N$ for several network architectures specified by the depth and the width for (a) the 2-local label, (b) the 3-local label, (c) 2-global label, and (d) 3-global label. Error bars indicate the standard deviation of the test error for 10 iterations of the network initialization and the training. Test errors calculated by the NTK of the depth of 1 and 7 are also plotted.

## 3.2 OPPOSITE DEPTH DEPENDENCES FOR $k$-LOCAL AND $k$-GLOBAL LABELS WITH $k \geq 2$

For $k \geq 2$, it turns out that experimental results show opposite depth dependences for $k$-local and $k$-global labels. Let us first consider $k$-local labels with $k \geq 2$. Figure 2 (a) and (b) show test errors for varying $N$ in various networks for the 2-local and the 3-local labels, respectively. The input dimension $d$ is set to be $d = 500$ in the 2-local label and $d = 100$ in the 3-local label. We see that the test error strongly depends on the network depth. The deeper network ($L = 7$) generalizes better than the shallower one ($L = 1$). It should be noted that for $d = 500$, the network of $L = 1$ and $H = 2000$ contains about $10^6$ trainable parameters, the number of which is much larger than that of trainable parameters ($\simeq 10^5$) in the network of $L = 7$ and $H = 100$. Nevertheless, the latter outperforms the former in the 2-local label as well as in the 3-local label with large $N$, which implies that increasing the number of trainable parameters does not necessarily imply better generalization. In $k$-local labels with $k \geq 2$, the network depth is more strongly correlated to generalization compared with the network width.

From Fig. 2 (b), it is obvious that the network of $L = 7$ and $H = 100$ fails to learn the 3-local label for small $N$. We also see that error bars of the test error are large in the network of $L = 7$ and $H = 100$. The error bar represents the variance due to initialization and training. By increasing the network width $H$, both variances and test errors decrease. This result is consistent with the recent observation in the lazy regime that increasing the network width results in better generalization because it reduces the variance due to initialization (D'Ascoli et al., 2020).

Next, we consider $k$-global labels with $k = 2$ and 3. The input dimension $d$ is set as $d = 100$ for the 2-global label and $d = 40$ for the 3-global label. We plot test errors against $N$ in Fig. 2 for (c) the 2-global label and (d) the 3-global label. Again we find strong depth dependences, but now shallow networks ($L = 1$) outperform deep ones ($L = 7$), contrary to the results for $k$-local labels. For $L = 7$, we also find strong width dependences; the test error of a wider network more quickly

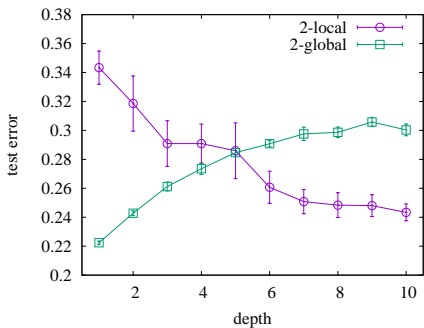

Figure 3: Depth dependence of the test error for $N = 10^4$ training samples with 2-local and 2-global labels. The dimension of input vectors is set to be $d = 500$ in the 2-local label and $d = 100$ in the 2-global label. The network width is fixed to be 500. An error bar indicates the standard deviation over 10 iterations of the training using the same dataset.

decreases with $N$. In particular, in the 3-global label, an improvement of the generalization with $N$ is minor for $L = 7$ and $H = 100$. An increase in the width makes a decrease in the test error with $N$ much faster [see the result for $L = 7$ and $H = 500$ in Fig. 2 (d)].

To see more details of the effect of depth, we also plot the depth dependence of the test error for fixed training data samples. We prepare $N = 10000$ training data samples for the 2-local and 2-global labels, respectively. The input dimension is $d = 500$ for the 2-local label and $d = 100$ for the 2-global label. By using the prepared training data samples, networks of the depth $L$ and the width $H = 500$ are trained up to $L = 10$. The test errors of trained networks are shown in Fig. 3. The test error of the 2-local label decreases with increasing $L$, whereas the test error of the 2-global label increases with $L$. Thus, Fig. 3 clearly shows the opposite depth dependences for local and global labels.

### 3.3 COMPARISON BETWEEN FINITE NETWORKS AND NTKS

In Figs. 1 and 2, test errors calculated by using the NTK are also plotted. In the case of $k = 1$ (Fig. 1) and the 2-global label [Fig. 2 (c)], the generalization performance of the NTK is comparable to or lower than that of finite networks.

A crucial difference is seen in the case of the $k$-local label with $k = 2$ and 3 and the 3-global label. In Fig. 2 (a) and (b), we see that the NTK almost completely fails to classify the data, although finite networks are successful. In the case of the 3-global label, the NTK of depth $L = 7$ correctly classifies the data, while the NTK of depth $L = 1$ fails [see Fig. 2 (d)]. In those cases, the test error calculated by a finite network does not seem to converge to that obtained by the NTK as the network width increases.

The NTK has been proposed as a theoretical tool to investigate the infinite-width limit. However, it should be kept in mind that the learning rate has to be sufficiently small to achieve the NTK limit (Jacot et al., 2018; Arora et al., 2019). The discrepancy between a wide network and the NTK in Fig. 2 stems from the strong learning-rate dependence of the generalization performance. In our experiment, the learning rate has been optimized by performing a 10-fold cross validation. If the optimized learning rate is not small enough for each width, the trained network may not be described by the NTK even in the infinite-width limit.

In Fig. 4 (a), we plot the learning-rate dependence of the test error for the 2-local label and the 2-global label in the network of the depth $L = 1$ and the width $H = 2000$. We observe a sharp learning-rate dependence in the case of the 2-local label in contrast to the case of the 2-global label. In Fig. 4 (b), we compare the learning-rate dependences of the test error for $L = 1$ and $L = 7$ in the case of the 3-global label ($H = 2000$ in both cases). We see that the learning-rate dependence for $L = 1$ is much stronger than that for $L = 7$, which is consistent with the fact that the NTK fails only for $L = 1$. It should be noted that Fig. 4 (b) shows that the deep network ($L = 7$) outperforms

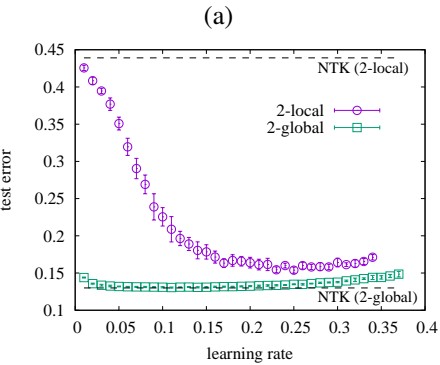
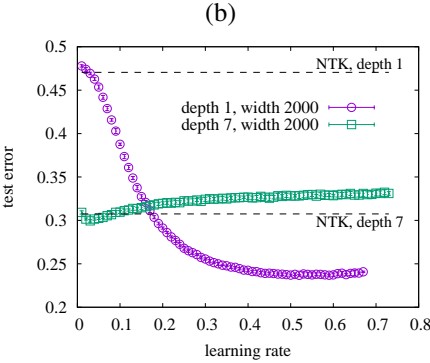

Figure 4: Learning-rate dependence of the test error. (a) Numerical results for the 2-local and 2-global labels in the network with the depth of 1 and the width of 2000. (b) Numerical results for the 3-global label in the networks with the depth of 1 and 7 (the network width is set at 2000 for both cases). The dotted lines show the test error calculated by the NTK. Each data is plotted up to the maximum learning rate beyond which the zero training error is not achieved within 2500 epochs (in some cases training fails due to divergence of network parameters during the training). Error bars indicate the standard deviation over 10 iterations of the training.

the shallow one ($L = 1$) in the regime of small learning rates, while the shallow one performs better than the deep one at their optimal learning rates.

Figure 4 also shows that the test error for a sufficiently small learning rate approaches the one obtained by the corresponding NTK. Therefore, the regime of small learning rates is identified as a lazy learning regime, while larger learning rates correspond to a feature learning regime. An important message of Fig. 4 is that we should investigate the feature learning regime, rather than the lazy learning regime, to correctly understand the effect of the depth on generalization.

## 4    DISCUSSION ON THE OBSERVED DEPTH DEPENDENCE

It is crucial for our problem to understand why deeper (shallower) is better for local (global) labels. As a possible mechanism, we suggest that local features can naturally be detected with the help of the chaotic signal propagation through a deep network (Poole et al., 2016) due to multiplicative growth of a small perturbation on an input $x$ across each layer.

An important distinction between the $k$-local label and the $k$-global label with $k \ll d$ lies in their stability against perturbations. We can typically change the $k$-local label of an input $x \in \mathbb{R}^d$ by moving $x$ to $\tilde{x} = x + z$ with $\|z\|_1 = O(1)$, where $z \in \mathbb{R}^d$ is a perturbation and $\| \cdot \|_1$ stands for the Manhattan distance (the $L^1$ distance). On the other hand, a stronger perturbation $\|z\|_1 = \tilde{O}(\sqrt{d})$ is typically needed to change the $k$-global label.

A recent study by De Palma et al. (2019) shows that a randomly initialized pre-trained wide neural network generates a label such that a perturbation $z$ with $\|z\|_1 = \tilde{O}(\sqrt{d})$ is typically needed to change it[3], indicating that $\tilde{O}(\sqrt{d})$ is a natural scale of resolution for a pre-trained random neural network. Compared with it, learning the $k$-local label requires finer resolution: two close inputs $x$ and $x'$ with $\|x - x'\|_1 = O(1)$ should result in distinct outputs $f(x)$ and $f(x')$ when $x$ and $x'$ have different labels. A deep neural network can naturally implement such a situation via chaotic signal propagation, and hence the depth is beneficial for learning local features. On the other hand, the $k$-global label is as stable as a typical label generated by a pre-trained network, and the chaoticity is not needed: two close inputs $x$ and $x'$ with $\|x - x'\|_1 \ll \tilde{O}(\sqrt{d})$ typically share the same label and therefore the outputs $f(x)$ and $f(x')$ should also be close to each other. In this case, the chaoticity may bring about unnecessary high resolution and therefore be rather disadvantageous: a shallower network is better for the $k$-global label. This is a likely explanation of the result obtained in Sec. 3.2.

---

[3]This statement has been proved for binary inputs with the Hamming distance, but we expect that it also holds for continuous inputs with the Manhattan distance.

Whereas the chaotic signal propagation was originally discussed to explain high expressivity of deep networks (Poole et al., 2016), we emphasize that the benefit of depth in local labels is not due to high expressivity since learning the $k$-local label with $k \ll d$ does not require high expressivity.[4] Nevertheless, the chaoticity of deep neural networks may play a pivotal role here.

To support the above intuitive argument, we experimentally show in Appendix B that a deeper network has a tendency towards learning more local features even in training *random labels*. That is, even for learning a completely structureless dataset, learned features in a deep network tend to be more local compared with those in a shallow one, which is possibly due to chaoticity of deep networks and potentially explains the result of Sec. 3.2.

## 5 CONCLUSION

In this work, we have studied the effect of increasing the depth in classification tasks. Instead of using real data, we have employed an abstract setting with random inputs and simple classification rules because such a simple setup helps us to understand under what situations deeper networks perform better or worse. We find that the locality of relevant features for a given classification rule plays a key role. In Sec. 4, we have proposed that the chaotic signal propagation in deep networks gives a possible mechanism to explain why deeper is better in local labels.

It is also an interesting observation that shallower networks do better than deeper ones for the $k$-global label, indicating that the depth is not necessarily beneficial. Although our dataset with the $k$-local or $k$-global label is artificial, there are some realistic examples in which $k$-global label is relevant, i.e., *thermodynamic systems in physics*. Thermodynamic variables in physics are usually expressed in the form of Eq. (2), i.e., a sum of local quantities over the entire volume of the system, and hence are regarded as global features. Our result therefore implies that a shallower network is more suitable for machine learning of thermodynamic systems. In Appendix C, we demonstrate that it is indeed the case in a classification problem of spin configurations of the Ising model, which is a quintessential model of ferromagnets Nishimori & Ortiz (2011). In typical image classification tasks, local features would be important and deeper networks do better, but we expect that there are some real datasets in which global features are more important, such as an example discussed in Appendix C.

We believe that our finding is relevant to real datasets, including the physics models discussed above. It is an important future problem to understand how to apply our findings to machine learning of real datasets.

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

# A  EXPLICIT EXPRESSION OF THE NTK

We consider a network whose biases $\{b^{(l)}\}$ and weights $\{w^{(l)}\}$ are randomly initialized as $b_i^{(l)} = \beta \tilde{b}_i^{(l)}$ with $\tilde{b}_i^{(l)} \sim \mathcal{N}(0, 1)$ and $w_{ij}^{(l)} = \sqrt{2/n_{l-1}} \tilde{w}_{ij}^{(l)}$ with $\tilde{w}_{ij}^{(l)} \sim \mathcal{N}(0, 1)$ for every $l$, where $n_l$ is the number of neurons in the $l$th layer, i.e., $n_0 = d, n_1 = n_2 = \cdots = n_L = H$. In the infinite-width limit $H \to \infty$, the pre-activation $f^{(l)} = w^{(l)} z^{(l-1)} + b^{(l)}$ at every hidden layer tends to an i.i.d. Gaussian process with covariance $\Sigma^{(l-1)} : \mathbb{R}^d \times \mathbb{R}^d \to \mathbb{R}$ that is defined recursively as

$$
\begin{cases}
\Sigma^{(0)}(x, x') = \dfrac{x^{\mathrm{T}} x'}{d} + \beta^2; \\
\Lambda^{(l)}(x, x') = \begin{pmatrix} \Sigma^{(l-1)}(x, x) & \Sigma^{(l-1)}(x, x') \\ \Sigma^{(l-1)}(x', x) & \Sigma^{(l-1)}(x', x') \end{pmatrix}; \\
\Sigma^{(l)}(x, x') = 2\mathbb{E}_{(u,v) \sim \mathcal{N}(0, \Lambda^{(l)})} [\varphi(u)\varphi(v)] + \beta^2
\end{cases}
\tag{12}
$$

for $l = 1, 2, \ldots, L$. We also define

$$
\dot{\Sigma}^{(l)}(x, x') = 2\mathbb{E}_{(u,v) \sim \mathcal{N}(0, \Lambda^{(l)})} [\dot{\varphi}(u)\dot{\varphi}(v)],
\tag{13}
$$

where $\dot{\varphi}$ is the derivative of $\varphi$. The NTK is then expressed as $\Theta_{ij}^{(L)}(x, x') = \delta_{i,j} \Theta^{(L)}(x, x')$, where

$$
\Theta^{(L)}(x, x') = \sum_{l=1}^{L+1} \left( \Sigma^{(l-1)}(x, x') \prod_{l'=l}^{L+1} \dot{\Sigma}^{(l')}(x, x') \right).
\tag{14}
$$

The derivation of this formula is given by Arora et al. (2019).

Using the ReLU activation function $\varphi(u) = \max\{u, 0\}$, we can further calculate $\Sigma^{(l)}(x, x')$ and $\dot{\Sigma}^{(l)}(x, x')$, obtaining

$$
\Sigma^{(l)}(x, x') = \frac{\sqrt{\det \Lambda^{(l)}}}{\pi} + \frac{\Sigma^{(l-1)}(x, x')}{\pi} \left[ \frac{\pi}{2} + \arctan\left( \frac{\Sigma^{(l-1)}(x, x')}{\sqrt{\det \Lambda^{(l)}}} \right) \right] + \beta^2
\tag{15}
$$

and

$$
\dot{\Sigma}^{(l)}(x, x') = \frac{1}{2} \left[ 1 + \frac{2}{\pi} \arctan\left( \frac{\Sigma^{(l-1)}(x, x')}{\sqrt{\det \Lambda^{(l)}}} \right) \right].
\tag{16}
$$

For $x = x'$, we obtain $\Sigma^{(l)}(x, x) = \Sigma^{(0)}(x, x) + l\beta^2 = \|x\|^2/d + (l+1)\beta^2$. By solving eqs. (15) and (16) iteratively, the NTK in Eq. (14) is obtained.[5]

# B  LOCAL STABILITY EXPERIMENT

Rather than considering general perturbations, we here restrict ourselves to local perturbations, $z = \pm v e^{(i)}$, where $v > 0$ is the strength of a perturbation and $e^{(i)}$ is the unit vector in $i$th direction ($i = 1, 2, \ldots, d$). For each sample $x'^{(\mu)}$ in the test data, if the network does not change its predicted label under any local perturbation for a given $v$, i.e., $\arg\max_{j \in [K]} f_j(x'^{(\mu)}) = \arg\max_{j \in [K]} [f_j(x'^{(\mu)} + \alpha v e^{(i)})]$ for any $i \in [d]$ and $\alpha \in \{\pm 1\}$, we say that the network is $v$-stable at $x'^{(\mu)}$. We define $s^{(\mu)}(v)$ as $s^{(\mu)}(v) = 1$ if the network is $v$-stable at $x'^{(\mu)}$ and $s^{(\mu)}(v) = 0$ otherwise. The local stability of the network is measured by a quantity

$$
s(v) = \frac{1}{N_{\text{test}}} \sum_{\mu=1}^{N_{\text{test}}} s^{(\mu)}(v).
\tag{17}
$$

---

[5]When $\beta = 0$ (no bias), the equations are further simplified; $\Sigma^{(l)} = \frac{\|x\|\|x'\|}{d} \cos\theta^{(l)}$ and $\dot{\Sigma}^{(l)} = 1 - \frac{\theta^{(l-1)}}{\pi}$, where $\theta^{(0)} \in [0, \pi]$ is the angle between $x$ and $x'$, and $\theta^{(l)}$ is iteratively determined by

$$
\cos\theta^{(l)} = \frac{1}{\pi} \left[ \sin\theta^{(l-1)} + (\pi - \theta^{(l-1)}) \cos\theta^{(l-1)} \right].
$$

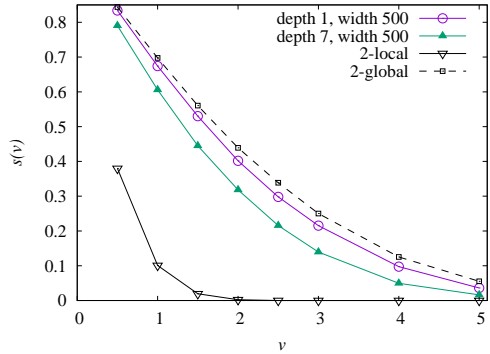

Figure 5: Local stability $s(v)$ for a shallow network ($L = 1$, $H = 500$), a deep network ($L = 7$, $H = 500$), the 2-local label, and the 2-global label. The networks are trained on training dataset with randomized labels. We can see that $s(v)$ for a deep network is smaller than that for a shallow one, which implies that a deeper network has a tendency to learning more local features even if the same dataset is given.

The $v$-stability can be defined similarly for the $k$-local and $k$-global labels: a given label is said to be $v$-stable at $x'^{(\mu)}$ if the label does not change under any local perturbation with strength $v$. We can also define $s(v)$ for $k$-local and $k$-global labels. In our setting, $v = O(1)$ is sufficient to change the $k$-local label, whereas typically $v = \tilde{O}(\sqrt{d})$ is needed to alter the $k$-global label.

We measure $s(v)$ for shallow ($L = 1$ and $H = 500$) and deep ($L = 7$ and $H = 500$) networks trained on $N = 10^4$ datasets with random labels by using the SGD of the learning rate $\eta = 0.1$ and the minibatch size $B = 50$. The dimension of input vectors is fixed as $d = 100$. In Fig. 5, $s(v)$ is plotted for the shallow and deep networks as well as for the 2-local and 2-global labels. We observe that $s(v)$ in the deep network is smaller than that in the shallow one. Moreover, $s(v)$ in the shallow network is close to that of the 2-global label. These experimental results indicate that even for learning a completely structureless dataset, learned features in a deep network tend to be more local compared with those in a shallow one.

## C  CLASSIFICATION OF ISING SPIN CONFIGURATIONS

In the main text, we have considered rather artificial datasets (i.i.d. random Gaussian inputs and $k$-local or $k$-global label). Our main result is that local (global) features are more efficiently learned by using deeper (shallower) networks. Here, we demonstrate that this result is relevant to a realistic dataset in physics, i.e., a dataset constructed from snapshots of Ising spin configurations.

The Ising model has been studied in physics as a quintessential model for ferromagnets (Nishimori & Ortiz, 2011). At each site $i = 1, 2, \ldots, d$, a binary spin variable $\sigma_i \in \{\pm 1\}$ is placed, and a spin configuration $x$ is specified by a set of spin variables, i.e., $x = (\sigma_1, \sigma_2, \ldots, \sigma_d)^{\mathrm{T}}$. In the one-dimensional Ising model at inverse temperature $\beta$, spin configurations are generated according to a Gibbs distribution,

$$P_\beta(x) = \frac{1}{Z_\beta} e^{-\beta H(x)}, \tag{18}$$

where $Z_\beta$ is a normalization constant known in statistical mechanics as the partition function and

$$H(x) = \sum_{i=1}^{d} \sigma_i \sigma_{i+1} \tag{19}$$

is called the Hamiltonian, which gives the energy of the system.

Now we fix two different inverse temperatures $\beta_1$ and $\beta_2$, and generate spin configurations following either $P_{\beta_1}(x)$ or $P_{\beta_2}(x)$ with equal probability by using a Markov-chain Monte Carlo method. Then we ask whether a given spin configuration $x$ is generated by $P_{\beta_1}(x)$ or $P_{\beta_2}(x)$. This is a classification problem, which can be solved by machine learning.

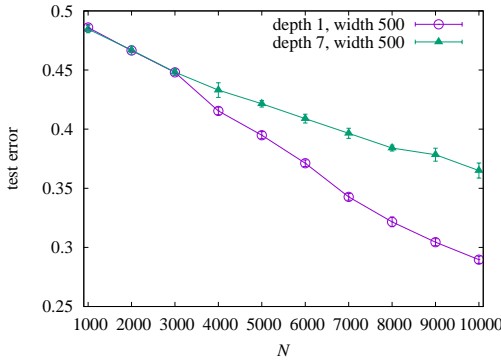

Figure 6: Test error against the number of training data samples $N$. A shallow network of depth 1 and width 500 shows better generalization compared with a deep network of depth 7 and width 500.

We consider supervised learning for this classification problem, where we fix $d = 500$, $\beta_1 = 0.1$, and $\beta_2 = 0.3$. A training dataset $\mathcal{D} = \{(x^{(\mu)}, y^{(\mu)}) : \mu = 1, 2, \ldots, N\}$ consists of spin configurations $\{x^{(\mu)}\}$ and their labels $\{y^{(\mu)}\}$. If $x^{(\mu)}$ is generated at the inverse temperature $\beta_1$ (or $\beta_2$), its label is given by $y^{(\mu)} = 1$ (or 2). Training is done with the same procedure as in Sec. 2.3. After training, we measure the test error by using an independently prepared test dataset with $10^5$ samples.

The temperature for a spin configuration $x$ is most successfully estimated by measuring its energy, i.e., the value of $H(x)$. Therefore $H(x)$ plays the role of a relevant feature in this classification problem. It is obvious from Eq. (19) that the energy is a 2-global quantity, and hence, this problem is similar to the classification of the dataset with the 2-global label. We then expect that, from our main result discussed in the main text, a shallower network will do better than a deeper one.

Experimental results for a shallow network ($L = 1$ and $H = 500$) and a deep one ($L = 7$ and $H = 500$) are shown in Fig. 6. For small $N$, both networks show almost identical performance, while for larger $N$, the shallow network clearly outperforms the deep one.

Apart from this example, thermodynamic quantities in physics are usually expressed in terms of a sum of local quantities over the entire volume of the system, and hence they are global variables. Our main result thus implies that shallower networks are more suitable for machine learning of thermodynamic systems.

## D    RESULTS FOR THE MEAN-SQUARE LOSS

We shall present experimental results for the mean-square loss. Figure 7 shows test errors for varying $N$ in various networks for (a) the 2-local label, (b) the 2-global label, (c) 3-local label, and (d) 3-global label. As for the depth dependences, we find qualitatively the same behavior as in the cross-entropy loss. That is, deeper is better for local labels, while shallower is better for global labels.

However, we also find that generalization performance quantitatively depends on the choice of the loss function. Overall, the cross-entropy loss yields better generalization than the mean-square loss. In particular, for the 2-local and 3-local labels, results for the mean-square loss tend to be closer to those in the NTK. It indicates that the network trained with the cross-entropy loss more easily gets out of the lazy learning regime compared with the network trained with the mean-square loss. This is natural since the weights in the network trained with the cross entropy loss will eventually go to infinity after all the training data samples are correctly classified (Lyu & Li, 2020) if no regularization such as the weight decay is used. On the other hand, for the mean-square loss, the network parameters can stay close to their initial values and therefore in the lazy learning regime throughout training Arora et al. (2019).

We also find that test errors for the mean-square loss show stronger dependences on the width of the network, especially for local labels. For example, let us compare Fig. 2 (b) and Fig. 7 (b).

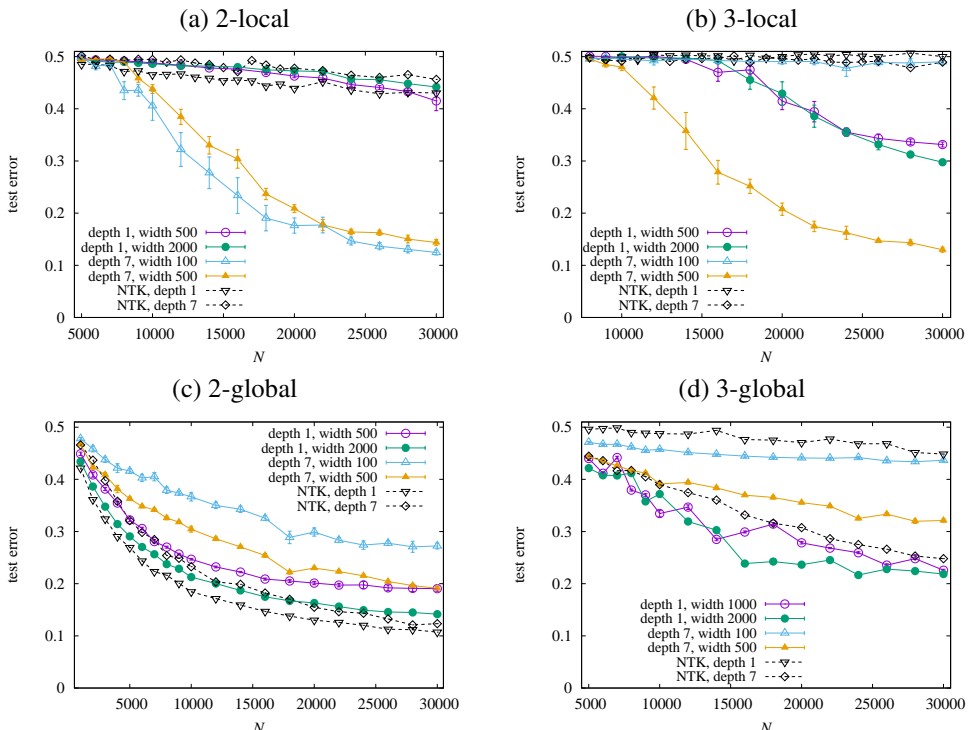

Figure 7: Test error against the number of training data samples $N$ for several networks trained with the mean-square loss for (a) the 2-local label, (b) the 3-local label, (c) 2-global label, and (d) 3-global label. Error bars indicate the standard deviation of the test error for 10 iterations of the network initialization and the training. Test errors calculated by the NTK of the depth of 1 and 7 are also plotted.

In the case of the cross-entropy loss, the network of $L = 7$ and $H = 100$ fails to generalize for small $N$ ($N \lesssim 15000$) but generalizes well for larger $N$. On the other hand, in the case of the mean-square loss, the same network does not generalize well at least up to $N = 30000$, although the network of $L = 7$ and $H = 500$ does. Strong width dependences found in Fig. 7 are consistent with recent rigorous results (Zou et al., 2020; Nitanda et al., 2019; Ji & Telgarsky, 2019; Chen et al., 2019), which show that the cross-entropy loss requires a much smaller width compared with the mean-square loss to realize an overparameterized regime.

