# OpenReview forum: "Is deeper better? It depends on locality of relevant features"
_ICLR.cc/2021/Conference — Reject_

### Official Review · AnonReviewer2 · 2020-10-19
**Clever artificial dataset with artificial notion of local vs. global features to study infinite NTK and finite networks performance.**

**Rating:** 7
**Confidence:** 4

**Review:**

This paper aims to empirically explore the depth dependence of overparameterized networks. The authors study fully-connected networks trained on a synthetic dataset consisting of random Gaussian inputs, with the label a simple function of the input. In one case, for "local" labels, the label is the parity of a product of a subset of the components. In the other case, for "global" labels, the label is a sum of such products of subsets with coverage over all the components. Broadly, the authors find that deeper MLPs are better able to learn the local labels, but shallower MLPs are better able to learn the global labels. Finally, the authors compare these results to the infinite width NTK and show that the NTK does not at all capture the behavior of the finite networks.

I think the strongest point of the paper is that the authors are able to find such a robust effect. The design of the synthetic dataset is very clever, and the results on depth and finite vs. infinite width are very compelling.

I think my biggest concern is whether this robust effect relates to global vs. local features in real data in the way that the authors suggest. I agree that the functions considered by the authors have a nice notion of "local" and "global" but I'm not sure how good a representation this is of the types of features that are present in real data. At the very least, it would have been nice to have some kind of discussion of the relationship, and at best the authors could have provided some kind of evidence that the features in an exemplar real dataset can be organized this way. I think that this is not completely obvious since the features considered here are multiplicative in components and the label is based on parity. (Do local features in images have this property? What do the authors have in mind for a global feature in an image? I highly doubt that things are arranged in sums of products over the whole image, with the parity of the result being important.) I understand that this is a toy model and I appreciate the nice result; I just would like to understand how representative this toy is.

I think that this is a good paper, and I recommend that ICLR accepts.

I think the experiments are clever, the results are robust, and -- despite my concerns about how to relate this dataset to "real data" -- the discrepancy between local and global information and depth is really interesting. I also find the comparison with the infinite width NTK to be very useful; it's really important to understand the ways in which infinite networks are good models for realistic finite networks, and here the authors demonstrate that such infinite networks completely fail on this task.

One question I had for the authors is why they think that shallow networks can learn global features better than deeper networks. I don't think the authors try to offer an explanation for this effect at any point.

Additionally, I was wondering whether the authors did any experiments for much larger k. Clearly k=d is not that interesting, but were there experiments in a middle regime rather than k << d?

Finally, could the authors comment on the relationship of their results to that of arxiv2007.15801? In that paper, for MLPs, the authors seem to find that infinite width networks broadly perform better than finite width networks. Inspection of your Figure 2(c) suggests one possibility is that real data is more akin to having global information? On the other hand, in that paper the authors find that finite networks outperform infinite networks for convolution networks (which obviously care about features that are local in space, and not the less-restrictive k-local notion of locality). Did the authors consider or make any experiments on convolutional architectures?

Did the authors consider any other initializations other than what's discussed in the paper and footnote 2?

A minor comment: I think footnote 1 is a little misleading. The "k-global" label type would still be considered "k-local" from a quantum information perspective. (Of course the authors are welcome to describe their invented dataset however they like.)

---

> ### Author Response · Authors · 2020-11-18
> **Response to Reviewer 2**
>
> We thank the reviewer for the insightful review. Below we reply to your comments/concerns.
>
> 1. *I think my biggest concern is whether this robust effect relates to global vs. local features in real data in the way that the authors suggest.*
>
> We also think that the relation to real dataset is quite important, and we agree with reviewer's comment "*I think that this is not completely obvious*". Since real datasets are too complicated, at this stage, we cannot establish a precise relationship between our "k-locality" and locality in a generic real dataset.
>
> On the other hand, we point out that there exist realistic problems in which k-global features are considered to be important, i.e., machine learning of thermodynamic systems in physics. Since thermodynamic variables (such as the energy) are usually written as a global sum of local quantities, they are regarded as global features. Our main result then suggests that shallower network is better for such a problem. We added Appendix C, in which we reported experimental results on a simple classification problem of the snapshots of the Ising-spin configurations according to their temperatures. We found that a shallow network is indeed better than a deep one. This additional result implies that our result has a connection to real data.
>
>
> 2. *One question I had for the authors is why they think that shallow networks can lean global features better than deeper networks. I don't think the authors try to offer an explanation for this effect at any point*
>
> Thanks for this comment. In the previous version, we did not provide any explanation at all. In our revised paper, we argued a possible explanation on why deeper is better for local labels and shallower is better for global labels in Section 4.
> We argued that the chaotic signal propagation through a deep network may play a key role. An important insight is that learning k-local label requires a finer scale of resolution than a typical scale of resolution realized by a randomly initialized pre-trained neural network.
> It means that two similar inputs should result in distinct outputs when the two inputs have different labels. A deep network can implement such a situation by utilizing chaoticity of signal propagation. This is our intuitive picture on why deeper is better for local features. On the other hand, the k-global label is much more stable under perturbations to an input, and hence the chaoticity brings about unnecessary high resolution and can be disadvantageous. That is why shallower is better for global labels.
> We also added a related additional experiment in Appendix B.
>
>
> 3. *Additionally, I was wondering whether the authors did any experiments for much larger k. Clearly k=d is not that interesting, but were there experiments in a middle regime rather than k << d?*
>
> We did experiments in a middle regime, but we found that a trained network on the k-local or k-global label with k comparable with d hardly generalizes for not too small d and N up to 50000. If N is further increased, the trained network might start to generalize, but due to its large computation cost, we did not try to access much larger N.
> Alternatively, we can consider a much smaller d, e.g. d=10 and k=5, but we think it is not so interesting.
>
>
> 4. *Finally, could the authors comment on the relationship of their results to that of arxiv2007.15801? In that paper, for MLPs, the authors seem to find that infinite width networks broadly perform better than finite width networks. Inspection of your Figure 2(c) suggests one possibility is that real data is more akin to having global information? On the other hand, in that paper the authors find that finite networks outperform infinite networks for convolution networks (which obviously care about features that are local in space, and not the less-restrictive k-local notion of locality). Did the authors consider or make any experiments on convolutional architectures?*
>
> In our setting, the NTK sometimes outperforms finite networks for the k-global label. Therefore, it might be plausible to consider that real dataset contains global features. However, real dataset will contain both local and global features and the comparison is not trivial, and it is difficult to give a definite interpretation.
>
> As for the convolutional architecture, we have not made any experiment so far since we wanted to first understand the behavior in the simplest setting. We think that it is an important future problem to investigate CNNs and understand the role played by the spatial locality in the data.
>
>
> 5. *Did the authors consider any other initializations other than what's discussed in the paper and footnote 2?*
>
> No, we have not tried other initializations.
>
> 6. *A minor comment: I think footnote 1 is a little misleading. The "k-global" label type would still be considered "k-local" from a quantum information perspective.*
>
> Thanks. We corrected the footnote 1.

---

### Official Review · AnonReviewer4 · 2020-10-27
**Insightful experiments, a little preliminary**

**Rating:** 6
**Confidence:** 3

**Review:**


This paper analyzes overparametrized networks evaluating how depth and width affect the generalization performance of the network. A set of experiments is designed in which labels are determined either by local or global interactions among the features, and generalization is observed for different values of width and depth of the network. NTK is also considered as a limit case of a network with infinite width.

The paper is well-written, and the experiments properly designed and explained. I appreciate the rigorous study, and the observations drawn, although theoretical arguments in support of the conclusion would have made them stronger. The connection between depth and local labels, and between shallowness and global label intuitively makes sense in the explanation of the authors, but I am left wondering how widely generalizable are these conclusions? What is the effect of the data generation procedure? Similarly the comparison with NTK is interesting, even if it seems to me more confirming NTK theory than providing clear support for the argument of the paper about the effects of depth; the conclusion of section 3.3 agrees with general NTK, but maybe it would have been helpful a longer discussion on how this relate to feature learning.

It seems to me that this work is good, but quite preliminary. It devised well-designed experiments, but all the contributions are empirical observations. It would definitely gain more value if a rigorous theoretical discussion of the results may be offered, or, if the same behavior may be observed on other, possibly real-world datasets.

---

> ### Author Response · Authors · 2020-11-18
> **Response to Reviewer 4**
>
> We thank the reviewer for the constructive review. We reply to the comments below.
>
> *The connection between depth and local labels, and between shallowness and global label intuitively makes sense in the explanation of the authors, but I am left wondering how widely generalizable are these conclusions? What is the effect of the data generation procedure?*
>
> As the reviewer2 also mentioned, it is not so trivial to connect our result with real dataset. However, in some cases, we can gain an insight from our result. In the revised version, we pointed out that there exist realistic problems in which k-global features are considered to be important, i.e., machine learning of thermodynamic systems in physics.
>
> Since thermodynamic variables (such as the energy) are usually written as a global sum of local quantities, they are regarded as global features. Our main result then suggests that shallower network is better for such a problem. We added Appendix C, in which we reported experimental results on a simple classification problem of the snapshots of the Ising-spin configurations according to their temperatures. We found that a shallow network is indeed better than a deep one. Applications of machine learning to physics has been an important topic of research, and this additional result implies that our result has a connection to real data.
>
>
> *Similarly the comparison with NTK is interesting, even if it seems to me more confirming NTK theory than providing clear support for the argument of the paper about the effects of depth; the conclusion of section 3.3 agrees with general NTK, but maybe it would have been helpful a longer discussion on how this relate to feature learning.*
>
> The main purpose of showing Figure 4 is not to show that a finite network performs similarly to the NTK in the small learning-rate regime, but to show that the feature learning regime (i.e. large learning rate regime) should be investigated to understand the effects of depth. In the revised version, we clearly state it at the end of section 3.3.
>
>
> *It would definitely gain more value if a rigorous theoretical discussion of the results may be offered, or, if the same behavior may be observed on other, possibly real-world datasets.*
>
> At present, we do not have a rigorous theoretical framework, but instead, we added a potential explanation on why deeper (shallower) is better for local (global) labels in section 4. We explained our empirical finding by the chaotic signal propagation through a deep neural network.
>
> As for real datasets, we point out that global features are important in physics datasets on thermodynamic systems, and our result offers a nontrivial prediction: shallower is better for such a dataset. In Appendix C, it is experimentally demonstrated that it is indeed the case.

---

### Official Review · AnonReviewer1 · 2020-10-28
**Highly controlled experiments that provide insight but confusing terminology**

**Rating:** 4
**Confidence:** 2

**Review:**

This paper proposes an empirical study of the nature of `"local" and "global" features and how well they can be approximated via fully connected networks under a highly controlled experimental setup.

Pros’:
The experiments are highly controlled and the research seems fairly reproducible
The paper is self-contained
The paper -- though containing a mathematical motivation linked to approximation theory -- is written in a very accessible way for non-expert researchers in the field (myself)
The overall question and motivation of the problem the authors are studying is of great importance in the fields of approximation theory, local vs global (Theory of Deep Learning, Applied Vision Science & Representation Learning) and learning capacity via SGD as well. [I am willing to raise my score if authors convince me what I have missed, or where I was confused -- see below].

Cons’:
There seems to be a misnomer on the definition of what local and global means. I am confused, and these two concepts that should be quite intuitive, are actually defined to mean different things (section 2.1 -- see critical observation below).
Authors present good empirical analysis but there is no potential explanation of their conclusion that suggests that global features can be approximated better with shallow networks, and local features with deeper networks. Arguments potentially linked to receptive field size and hierarchy like in vision science (Neyshabur, 2020; Deza, Liao, Banburki & Poggio 2020) would be interesting, but naturally authors cannot make such claims given that all networks trained/tested were fully connected with a R.F. size of 1.

----

Critical Observation [Section 2.1]

K-local label vs K-global label: Why does each k-local label depend on a multiplication of all k entries, while k-global depends on a sum of all k-entries? There seems to be a misnomer on what local and global means here. It is as if in both cases the label is determined by “global” properties (as all the inputs are used to compute the label), but the only difference is the operation type: multiplication vs sum -- but I could have missed something. This is quite confusing even though the authors state in the paper that locality need not mean spatial locality.

The problem with the confusing definition of locality and global properties is that it permeates into the introduction and conclusion of the paper and other relevant work. For example the claim in the introduction: “deeper is better for local labels, while shallow is better for global labels” under this new definition is confusing as it suggests something that is not what it seems. Unless local here literally means spatially local or implies that a function ignores/zeroes out other inputs (as it would be for functions with restricted -- hence local -- domains to compute the output), then the claim of the paper is misleading/confusing with the relevant literature. Authors should do a better job in arguing why they choose multiplication as a proxy for a local operation and summation as a proxy for a global one. If anything it would make more sense that a k-local label is essentially a subset of the k-global label (example a partial sum, vs a different operation altogether).

Again, at a higher level: I would have expected a sort of definition where there is only a single “global” label (vs k-global label) computed from a feature vector, and `multiple’ k-local labels, depending on what entries in the feature vector are sampled, such that as k reaches the total input size k-local is equal to “global” -- but this does not seem to be the case.

---
Sections 2.2 and onwards seem more clear.

Other observations: I really like the dissociation for k-local (continuing with the definition proposed by authors) and k-global labels expressed in Figure 2. This is a neat result, but again I think the argument made should be pushed towards the fundamentally different nature of the operation (multiplication vs addition -- and not local vs global)

The rest of the paper Figure 3 seems quite interesting to find the point in depth such that there is equalized performance for the 2-local and 2-global cases. On the other hand, I am not sure what Figure 4 is supposed to bring to the table (it seems like comparisons to performance with NTK) -- but, so what? Why does this matter?

---

I think the paper overall proposes a good first step via empirical analysis of the local vs global problem (under the authors definitions), but I would have wished that the paper ended on a higher note with regards to, *why* is k-locality approximated better with deeper networks, and k-global labels with shallow ones. Not that a proof would be necessary, but at least an intuition given the mathematical structure of the network (stacked dot products + non-linearity), their approximation power, and the nature of the label (multiplication vs addition).

---

> ### Author Response · Authors · 2020-11-18
> **Response to Reviewer 1**
>
> We thank the reviewer for the detailed review.
>
> First of all, we want to respond to your concern about the terminology of "local" and "global".
>
> The k-local label is determined by the sign of a product of k entries among d compoenents, whereas the k-global label is determined by a sum of such products over all the components.
> The label is determined by a small subset of the components in the k-local label (we assume that k is much smaller than d), whereas all the components contribute to the label in the k-global label.
>
> As the reviewer wrote, local here implies a function with a restricted domain to compute the output, and hence this definition of locality is quite intuitive.
> The difference between local and global labels is not the operation type.
>
> We hope that the above explanation clarifies reviewer's concerns about the terminology.
>
>
> Next, we want to reply to your comment:
>
> *On the other hand, I am not sure what Figure 4 is supposed to bring to the table*
>
> As the reviewer pointed out, we have to admit that the meaning of Figure 4 was not well explained in the previous version. Figure 4 is important because it shows the necessity to investigate the feature learning regime (large learning-rate regime) to understand the benefit of depth in learning local features. Although our understanding on the lazy-learning regime has advanced owing to recent important works, the feature learning regime remains less well understood. Our result would give some hint to gain more insights into the feature learning regime.
>
>
> Finally, we want to address your following comments:
>
> *Authors present good empirical analysis but there is no potential explanation of their conclusion that suggests that global features can be approximated better with shallow networks, and local features with deep networks.*
> *... but I have wished that the paper ended on a higher note with regards to, why is k-locality approximated better with deeper networks, and k-global labels with shallow ones*
>
> In our revised paper, we added a new section "Discussion on the observed depth dependence", where we gave a potential explanation on our result. We argued that the chaotic signal propagation through a deep neural network may play a key role. Learning the k-local label requires a relatively high resolution in the sense that the k-local label easily changes by adding local perturbations. It means that similar inputs must result in distinct outputs, which is naturally implemented by utilizing chaotic property of deep signal propagation. On the other hand, the k-global label is much more stable under local perturbations to an input, and hence the chaoticity due to depth can be rather disadvantageous. That is why shallower is better for the k-global label.
> We also performed a new experiment that is related to the above argument, and reported its result in Appendix B.

---

### Official Review · AnonReviewer3 · 2020-10-29
**Important topic, but the conclusions are not convincing**

**Rating:** 4
**Confidence:** 4

**Review:**

Recent theoretical study on the training of neural networks has introduced an important kernel function called neural tangent kernel. This paper studies the training of deep ReLU networks and compares it with the training directly using NTK by conducting experiments on synthetic data. Based on the experimental results, the authors conclude that deeper networks perform better on certain datasets whose labels are more “local”, while shallower networks are better at more “global” labels. Moreover, the authors observed that finite-width networks have better generalization than NTK.

I am not convinced by the claims of this paper, for the following reasons:

1. First of all, the definition of NTK in this paper in equation (10) may be wrong. If I understand it correctly, the definition of NTK in Jacot et al., 2018 uses gradient with respect to the W’s instead of the w’s. Therefore by chain rule, the definition in this paper differs from the correct definition by a large factor. This can be essential and may be the reason why the experimental performance of NTK is so bad. For the same reason, it may also be possible that not all layers of the network has been trained with the same importance (some layers change faster than other layers due to having a different chain-rule factor), and this can potentially lead to performance differences for NNs and NTKs with different depths.

2. Moreover, the comparison done in this paper is between finite neural networks trained with *cross-entropy loss* and NTK trained with *mean square loss*. The authors commented that replacing cross-entropy loss with mean square loss does not make much difference. However, in the NTK literature there is indeed some difference in landscape properties, over-parameterization requirement, etc (Zou et al., (arXiv:1811.08888), Nitanda et al., (arXiv:1905.09870), Ji & Telgarsky, (arXiv:1909.12292), Chen et al., (arXiv:1911.12360)). Moreover, for classification losses, if the network is trained for a very long time, it will eventually escape from the NTK regime since cross-entropy encourages the weights to go to infinity after all training samples are correctly classified (Lyu & Li, (arXiv:1906.05890)), while the whole training path for square loss is in the NTK regime (Arora et al., (arXiv:1904.11955)). Therefore I suggest that the authors should add training results for finite-width NN using square loss as well, and comment on how their experimental results reflect the results in the references mentioned above.

3. My third concern is closely related to the second one above. Throughout the paper, the authors discussed finite width NNs as if they are not in the NTK regime or lazy training regime (for example the 4th paragraph on page 2). This is not true, as almost all results in NTK regimes indeed study wide, but finite NNs. In fact, it is shown in Ji & Telgarsky, (arXiv:1909.12292) and Chen et al., (arXiv:1911.12360) that NNs with (almost) constant width can still fall in the NTK regime with classification losses (unless, of course, when trained for too long). It has been studied that whether the training is in NTK regime is not determined by the width, but the scaling of the network (Chizat et al., (arXiv:1812.07956), Mei et al., (arXiv:1902.06015), Chen et al., (arXiv:2002.04026)).

Because of the above two points 2 and 3, I am particularly skeptical about the authors’ explanation of the performance difference between finite NN and NTK.

4. There have been some slight differences in the definition of NTK. Particularly, it is discussed in Cao and Gu, (arXiv:1905.13210) that different definitions of NTK may differ in a 2^L factor for ReLU networks. I briefly checked Section A of the submission and it seems that the calculation of this paper matches Cao and Gu, (arXiv:1905.13210). Since the network depth is the focus of this paper, the authors may consider clarifying it.

---

> ### Author Response · Authors · 2020-11-18
> **Response to Reviewer 3**
>
> We thank the reviewer for carefully reading our paper and pointing out some important aspects of the NTK. We try to address reviewer's concerns below.
>
> 1. *First of all, the definition of NTK in this paper in equation (10) may be wrong. If I understand it correctly, the definition of NTK in Jacot et al., 2018 uses gradient with respect to the W’s instead of the w’s. Therefore by chain rule, the definition in this paper differs from the correct definition by a large factor. This can be essential and may be the reason why the experimental performance of NTK is so bad.*
>
> We thank the reviewer for pointing out that equation (10) in the previous version was wrong. We agree that gradients should be taken with respect to the scaled weights ($\tilde{w}$ in the revised version), so we modified Section 2.4. However, we did not use equation (10) directly for our experiments on the NTK, and this modification does not affect our numerical results. Experiments are properly performed by using correct formulae summarized in Appendix A.
>
>
> 2. (1) *Moreover, the comparison done in this paper is between finite neural networks trained with cross-entropy loss and NTK trained with mean square loss. The authors commented that replacing cross-entropy loss with mean square loss does not make much difference. However, in the NTK literature there is indeed some difference in landscape properties, over-parameterization requirement, etc (Zou et al., (arXiv:1811.08888), Nitanda et al., (arXiv:1905.09870), Ji & Telgarsky, (arXiv:1909.12292), Chen et al., (arXiv:1911.12360)).*
>
> We thank the reviewer for pointing out the importance of the choice of the loss function. In our paper, we commented that the mean-square loss yields similar results, but what we wanted to indicate is that qualitative results (deeper is better for local labels, whereas shallower is better for global labels) do not change by replacing the cross-entropy loss by the mean-square loss. Quantitatively, we actually observed the difference. Therefore, our result is consistent with the papers the reviewer cited.
>
>
> 2. (2) *Moreover, for classification losses, if the network is trained for a very long time, it will eventually escape from the NTK regime since cross-entropy encourages the weights to go to infinity after all training samples are correctly classified (Lyu & Li, (arXiv:1906.05890)), while the whole training path for square loss is in the NTK regime (Arora et al., (arXiv:1904.11955)). Therefore I suggest that the authors should add training results for finite-width NN using square loss as well, and comment on how their experimental results reflect the results in the references mentioned above.*
>
> According to reviewer's suggestion, we are now preparing experimental results using the mean-square loss. When all the numerical results are obtained, we will add them to our paper and make a brief comment on the relation to the references the reviewer pointed out.
>
>
> 3. *My third concern is closely related to the second one above. Throughout the paper, the authors discussed finite width NNs as if they are not in the NTK regime or lazy training regime (for example the 4th paragraph on page 2). This is not true, as almost all results in NTK regimes indeed study wide, but finite NNs. In fact, it is shown in Ji & Telgarsky, (arXiv:1909.12292) and Chen et al., (arXiv:1911.12360) that NNs with (almost) constant
> width can still fall in the NTK regime with classification losses (unless, of course, when trained for too long). It has been studied that whether the training is in NTK regime is not determined by the width, but the scaling of the network (Chizat et al., (arXiv:1812.07956), Mei et al., (arXiv:1902.06015), Chen et al., (arXiv:2002.04026)).*
>
> We agree that wide but finite networks can fall in the NTK regime when the learning rate is small enough. Indeed, in Figure 4, we find that the performance of a finite network is similar to that of the NTK in the small learning-rate regime. In order to avoid a misleading statement, we added a sentence "it is known that a wide but finite network can still be in the lazy learning regime for sufficiently small learning rates" in the last paragraph of section 1.
>
>
> 4. *There have been some slight differences in the definition of NTK. Particularly, it is discussed in Cao and Gu, (arXiv:1905.13210) that different definitions of NTK may differ in a 2^L factor for ReLU networks. I briefly checked Section A of the submission and it seems that the calculation of this paper matches Cao and Gu, (arXiv:1905.13210). Since the network depth is the focus of this paper, the authors may consider clarifying it.*
>
> We thank the reviewer for this comment. At the beginning of section 2.4 in our revised paper, we mentioned that our formulation follows Cao and Gu (and Arora et al.)

---

> > ### Author Response · Authors · 2020-11-24
> > **Mean-square loss**
> >
> > In the revised paper, we added experimental results for the mean-square loss and commented on how our experimental results reflect the results in the references mentioned by the reviewer.
> >
> > Our main result does not change, but we found quantitative difference in test errors. Overall, the cross-entropy loss yields better performance than the mean-square loss. We also found that test errors for finite networks tend to be closer to those for the NTK, compared with those for the mean-square loss. This indicates that a network trained with the cross-entropy loss more easily gets out of the lazy learning regime compared with that with the mean-square loss, which is consistent with the fact that "cross-entropy encourages the weights to go to infinity after all training samples are correctly classified (Lyu & Li, (arXiv:1906.05890)), while the whole training path for square loss is in the NTK regime (Arora et al., (arXiv:1904.11955))" as mentioned by the reviewer. However, we would like to emphasize that even for the mean-square loss, finite networks do better for local labels unless the width is too small.
> >
> > We also found that test errors for the mean-square loss show stronger dependences on the width of the network. This result is consistent with rigorous results obtained in Zou et al., (arXiv:1811.08888), Nitanda et al., (arXiv:1905.09870), Ji & Telgarsky, (arXiv:1909.12292), and Chen et al., (arXiv:1911.12360).
> >
> > We thank the reviewer for insightful comments and useful information, which help us to improve our paper.

---

### Author Response · Authors · 2020-11-18
**To all the reviewers**

We thank the reviewers for their constructive comments. We revised our paper according to reviewers' comments. According to the suggestion by the reviewer3, we are now preparing experimental results for the mean-square loss. Although we have not yet completed this job, at this stage we would like to upload the revised paper and reply to each reviewer.

The major change in this revision is summarized below:

* We added a new section 4, in which we give a potential explanation of our empirical finding, i.e. why deeper is better for local labels and shallower is better for global labels. We explained our result via the chaotic signal propagation through a deep neural network.

* We added Appendix B, in which an additional experimental result on the local stability of learned features with random labels is demonstrated, which supplements an intuitive argument in section 4. This experimental result suggests that a deeper network tends to learn more local features even for a completely structureless dataset.

* We added Appendix C, in which we show that our result is relevant for a real dataset: data generated by thermodynamic systems in physics. Since a thermodynamic quantity is regarded as a global feature, our empirical result implies that a shallow network is better than a deep one. We demonstrated that it is indeed the case.

* According to the comment by the reviewer3, we corrected equation (10) and a paragraph above it.

* According to the comment by the reviewer3, we mentioned in the last paragraph of section 1 that a wide but finite network can be in the lazy learning regime for sufficiently small learning rates.

---

> ### Author Response · Authors · 2020-11-24
> **Further revision: Results for the mean-square loss**
>
> We further revised our paper.
> * We added experimental results for the mean-square loss in Appendix D, according to the suggestion by the reviewer 3. The main result does not change (deeper is better for local labels and shallower is better for global labels), but we find quantitative differences in test errors for the mean-square loss and the cross entropy loss. We also discussed the relation to some previous studies mentioned by the reviewer 3.
> * We realized that not the Euclidean distance but the Manhattan distance should be used in section 4, so we fixed it. The reason is as follows: In the previous work by De Palma et al. (NeurIPS 2019, arXiv:1812.10156) on typical functions generated by random neural networks, binary inputs and the Hamming distance are used. In continuous inputs, the Hamming distance should be replaced by the Manhattan distance since the latter naturally reduces to the former for binary inputs. On the other hand, the Euclidean distance does not reduce to the Hamming distance for binary inputs.

---

### Decision · Program_Chairs · 2021-01-07
**Final Decision**

**Decision:**

Reject

**Comment:**

Reviewers found the construction is very clever and the empirical results are interesting. However, a more thorough theoretical explanation is needed for acceptance.